# LGR: LOCAL GEOMETRIC REFINEMENT IN HIGH-FIDELITY SURGICAL SCENE RECONSTRUCTION

## ABSTRACT

Dynamic reconstruction of deformable surgical scenes has the potential to significantly advance robot-assisted surgery. Building on recent advancements in 3D Gaussian splatting (3DGS), current surgical scene reconstruction (SSR) methods have made notable initial progress. Despite this progress, challenges remain in accurately tracking local tissue deformations during surgery, primarily due to the lack of deformation constraints within the local Gaussian neighborhoods of surgical tissues. In this work, we address these issues by proposing a local geometric refinement (LGR) framework based on 3DGS for high-fidelity SSR. Specifically, we first utilize prior visual information to efficiently perform the Gaussian initialization. Following the initialization, we incorporate local geometric constraints to accurately track the local non-rigid deformations occurring in the surgical scene. Furthermore, considering the low-quality scenarios in real surgeries, we apply low-quality enhancement to optimize the fidelity of local details in the preliminarily rendered scene. Experimental results on public datasets demonstrate that LGR outperforms previous state-of-the-art methods. Notably, it achieves an average improvement of over 50% in terms of LPIPS, a metric that better reflects human perceptual consistency, while maintaining favorable computational cost. These results highlight the great potential of the proposed LGR for promoting practical applications in surgical scenarios. *Our code and model will be released publicly.*

## 1 INTRODUCTION

Surgical scene reconstruction (SSR) plays a critical role in minimally invasive surgery (Yang et al., 2024c; Liu et al., 2024a; Long et al., 2021; Xie et al., 2024), enhancing the surgeon's understanding of the operative field and supporting various clinical applications, such as surgical simulation (Chong et al., 2022; Montaña-Brown et al., 2023), robotic surgery automation (Lu et al., 2021; Li et al., 2025), and medical education (Schmidt et al., 2024; Hashimoto et al., 2024). Traditional SLAM-based methods (Song et al., 2017; Zhou & Jagadeesan, 2019; Zhou & Jayender, 2021) struggle to address challenges posed by sparse viewpoints, dynamic scene deformations, and instrument occlusions (Gunderson et al., 2025; Yang et al., 2025; Wang et al., 2025; Yang et al., 2024b).

Recently, methods based on Neural Radiance Fields (NeRF) (Zha et al., 2023; Wang et al., 2022; Chen et al., 2025; Han et al., 2025; Gerats et al., 2024) have made initial progress in dynamic scene modeling for surgery. For example, EndoNeRF (Wang et al., 2022) and EndoSurf (Zha et al., 2023) enhance scene reconstruction by incorporating occlusion-aware modeling and joint shape–appearance representation strategies. However, the implicit representation of NeRF requires dense sampling of millions of rays to represent surgical scenes. Consequently, it incurs high computational costs when processing complex scenarios, thereby limiting its potential for real-time rendering (Xu et al., 2024; Yang et al., 2024c). To address the issue of inefficient rendering, methods based on 3D Gaussian Splatting (3DGS) (Kerbl et al., 2023) have been proposed (Zhu et al., 2024; Xie et al., 2024; Chen et al., 2024; Liu et al., 2024b; Yang et al., 2024b). These methods represent the scene as a series of 3D Gaussian distributions and render 2D images through the splatting-based rasterization process after tracking dynamic deformations. For instance, SurgicalGS (Chen et al., 2024) enhances reconstruction accuracy by using surgical motion masks, SurgicalGaussian (Xie et al., 2024) learns soft tissue deformations through multilayer perceptions (MLP), achieving higher quality scene renderings. Nevertheless, applying the aforementioned methods to the reconstruction of real dynamic surgical scenes still faces challenges: **i)** As depicted in the left image of Figure 1 **C)**, dynamic deformations

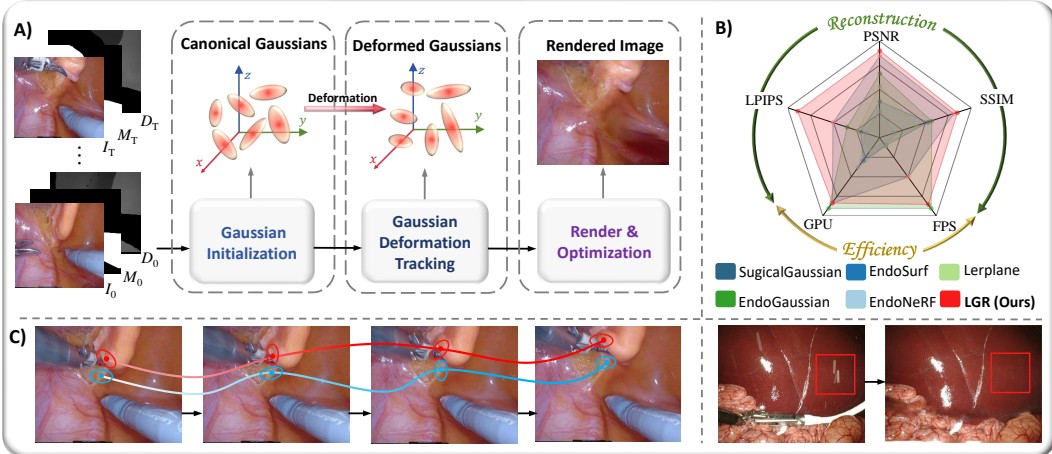

Figure 1: **A)** Illustration of the main workflow: Gaussian Initialization (GI), Gaussian Deformation Tracking (GDT), and Render & Optimization (RO). **B)** Radar chart of performance metrics: The proposed LGR achieves the best results on the EndoNeRF-Pulling dataset in terms of rendering quality ( PSNR ↑, SSIM ↑, LPIPS ↓ ) and shows certain advantages in rendering speed ( FPS ↑ ) and GPU ↓ memory usage ( ***Note:*** For clarity, LPIPS and GPU axes are visualized as $1 - $ LPIPS and $24 - $ GPU, respectively ). **C)** Two key points considered: the left image shows an example of local dynamic deformation, and the right image shows a visual example of low-quality reconstruction.

of local tissues occur in the surgical scene. Existing methods typically combine a single MLP (Zhu et al., 2024; Yang et al., 2024b; Huang et al., 2024; Liu et al., 2024b) with Gaussian point cloud rendering for global Gaussian deformation tracking, but do not adequately consider the geometric constraints of the local tissue Gaussian neighborhood, leading to insufficient tracking of complex and subtle local dynamic deformations. **ii)** As shown in the right image of Figure 1 **C)**, surgical scenes often contain low-quality interference such as splashes (Wu et al., 2024b;a), previous studies have not considered low-quality enhancement for rendering results.

To address these challenges, we propose a local geometric refinement (LGR) framework aimed at reconstructing high-fidelity models and enabling real-time rendering for SSR. As shown in Figure 1 **A)**, LGR is built upon 3DGS, with its workflow consisting of three main stages: Gaussian Initialization (GI), Gaussian Deformation Tracking (GDT), and Rendering & Optimization (RO). Specifically, as shown in Figure 2, in the GI stage, LGR rapidly performs Gaussian point cloud initialization for the surgical scene using input visual prior information (*i.e.*, RGB images, depth images, and tool masks). After initialization, the GDT stage utilizes a multi-head attribute decoder to capture variations in the attributes of the Gaussian points. Simultaneously, we design local geometric constraints (LGC) to constrain the positions, covariance, and feature consistency of the sampled Gaussian points and their neighboring points, further enhancing the alignment precision of Gaussian points in spatio-temporal space. To further enhance the scene detail reconstruction, LGR integrates a Low-quality Enhancement (LQE) module in the RO stage to address low-quality interference issues in real surgical scenes. As shown in the radar chart in Figure 1 **B)**, LGR outperforms existing methods across all rendering quality metrics on the EndoNeRF-Pulling dataset while maintaining low computational overhead. More experimental comparisons can be found in Section 4, Appendix A.4, and supplementary materials.

LGR has several appealing merits: **First, integrating local geometric constraints**: LGR introduces a local geometric constraint mechanism during Gaussian deformation tracking, which jointly constrains the spatial positions, covariance, as well as feature consistency of Gaussian points and their local neighborhoods. This effectively compensates for the loss of fine details caused by relying solely on global Gaussian optimization, thereby enabling higher-fidelity scene reconstruction. **Second, considering low-quality enhancement**: LGR further refines image quality after Gaussian reconstruction through low-quality enhancement, effectively mitigates the impact of low-quality data, such as splashes, commonly encountered in real surgical scenes. **Third, demonstrating practical applicability potential**: LGR consistently achieves accurate reconstruction results (particularly with an average improvement of over **50%** in LPIPS), better aligning with human visual perception, while maintaining low computational overhead, making it promising for deployment in medical scenarios.

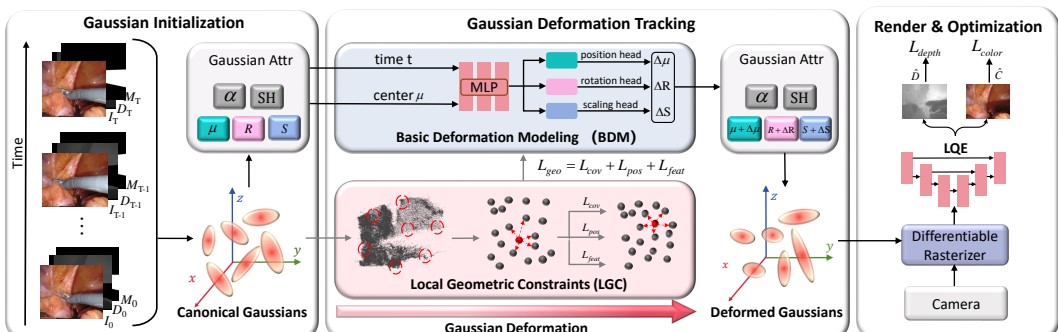

Figure 2: **Overview of LGR.** LGR comprises three stages: Gaussian Initialization (GI) (Sec. 3.2), Gaussian Deformation Tracking (GDT) (Sec. 3.3), and Render & Optimization (RO) (Sec. 3.4). In GI, a point cloud is initialized by back-projecting the rgb images, depth maps, and surgical tool masks to construct 3D Gaussians representing the canonical space. GDT decouples deformation modeling into Basic Deformation Modeling (BDM) and Local Geometric Constraints (LGC), where LGC is enforced via position ($L_{pos}$), covariance ($L_{cov}$), and feature ($L_{feat}$) losses, forming $L_{geo} = L_{cov} + L_{pos} + L_{fea}$. In RO, rgb and depth maps are rendered through a differentiable rasterizer, and after low-quality enhancement (LQE), compared with the inputs to compute $L_{color}$ and $L_{depth}$.

## 2 RELATED WORK

**NeRF for Surgical Scene Reconstruction.** Neural implicit representation (Mildenhall et al., 2021) has gained significant attention in medical imaging (Molaei et al., 2023; Feng et al., 2025; Shan et al., 2024; Wang et al., 2024; Choi et al., 2025). Neural Radiance Fields (NeRF), a representative of neural implicit representation, advances medical imaging by mapping input coordinates to corresponding values within the domain through implicit representation. Recent studies have extended neural implicit representations to dynamic deformable surgical scenes. EndoNeRF (Wang et al., 2022) first applies NeRF to deformable surgical scenes by integrating tool-guided ray casting, stereo-guided ray marching, and depth supervision, achieving effective reconstruction. Building on EndoNeRF (Wang et al., 2022), EndoSurf (Zha et al., 2023) enhances surface reconstruction by regularizing geometry with a signed distance field. However, this method requires optimizing the entire spatiotemporal field, which incurs high computational costs and limits its practical use in medical settings. To improve efficiency, subsequent works such as Lerplane (Yang et al., 2023) and Forplane (Yang et al., 2024a) decompose scene representation into 2D planes for static and dynamic components, speeding up optimization. Despite this, NeRF still necessitates dense sampling and querying of millions of rays, which inevitably leads to significantly increased computational overhead and decreased rendering speed, thereby limiting its practicality in medical applications (Liu et al., 2024b; Xie et al., 2024; Wang et al., 2025; Guo et al., 2025).

**3DGS for Surgical Scene Reconstruction.** 3D Gaussian Splatting (3DGS) (Kerbl et al., 2023) has demonstrated outstanding performance in modeling static scenes. By leveraging differentiable splat-based rendering with tile rasterization, it enables efficient scene reconstruction and rendering. Benefiting from these advantages, recent research has begun to adopt 3D Gaussian representations combined with deformation fields to model deformable surgical scenes (Liu et al., 2024a; Zhu et al., 2024; Xie et al., 2024; Li et al., 2024; Huang et al., 2024; Liu et al., 2024b). To handle dynamic tissue deformations of surgical organs, 3D Gaussians are typically coupled with deformation fields modeled in various ways. EndoSparse (Li et al., 2024) and SurgicalGaussian (Xie et al., 2024) employ MLPs to capture scene deformations, following a strategy similar to EndoNeRF (Wang et al., 2022). Another approach, adopted by Endo-GS (Zhu et al., 2024) and Endo-4DGS (Huang et al., 2024), models soft tissue deformation by combining multiple orthogonal 2D feature planes with a small MLP, inspired by Lerplane (Yang et al., 2023) and Forplane (Yang et al., 2024a), which further reduces training time. EndoGaussian (Liu et al., 2024b), on the other hand, adopts a motion-aware frame synthesis strategy to achieve high-fidelity reconstruction quality. However, most of these methods overlook the local fine-grained deformations in surgical scenes and the challenges associated with low-quality enhancement. In this paper, we combine visual prior information, local geometric constraints, and low-quality enhancement strategies to further improve reconstruction quality and rendering efficiency.

# 3 METHOD

## 3.1 PRELIMINARIES

We implement SSR using 3DGS as the underlying scene representation. As an explicit representation, 3DGS models the scene with a set of Gaussian primitives. Each Gaussian point has learnable attributes, including position $\boldsymbol{\mu} \in \mathbb{R}^3$, rotation $\boldsymbol{r} \in \mathbb{R}^4$, scale $\boldsymbol{s} \in \mathbb{R}^3$, opacity $\alpha$, and spherical harmonic (SH) coefficients for view-dependent appearance modeling. The spatial influence of each Gaussian is further parameterized by its mean position vector $\boldsymbol{\mu}$ and full covariance matrix $\boldsymbol{\Sigma} \in \mathbb{R}^{3 \times 3}$:

$$\boldsymbol{\Sigma} = \mathbf{R}\mathbf{S}\mathbf{S}^\top \mathbf{R}^\top, \quad G(x) = \exp\left(-\frac{1}{2}(x-\mu)^\top \boldsymbol{\Sigma}^{-1}(x-\mu)\right), \tag{1}$$

where $x \in \mathbb{R}^3$ denotes an arbitrary 3D coordinate in the world frame, and $\boldsymbol{\Sigma}$ is factorized into a scaling matrix $\mathbf{S}$ and a rotation matrix $\mathbf{R}$ for spatial transformation. Specifically, $\mathbf{R}$ is computed from rotation vector $\mathbf{r}$, and $\mathbf{S}$ is a diagonal matrix constructed from scale vector $\mathbf{s}$ to control anisotropic spread. Then, the color $\hat{C}(\mathbf{p})$ and depth $\hat{D}(\mathbf{p})$ of pixel $\mathbf{p}$ can be rendered using the following function:

$$\hat{\mathbf{C}}(\mathbf{p}) = \sum_{i=1}^{n} c_i \alpha_i \prod_{j=1}^{i-1}(1-\alpha_j), \quad \hat{\mathbf{D}}(\mathbf{p}) = \sum_{i=1}^{n} d_i \alpha_i \prod_{j=1}^{i-1}(1-\alpha_j), \tag{2}$$

where $c_i$ is the color computed from the SH coefficients of the $i$-th Gaussian, and $\alpha_i$ is obtained by evaluating the 2D covariance matrix $\boldsymbol{\Sigma}'_i$ and multiplying it by the learnable opacity value $o_i$. The 2D covariance matrix is computed as $\boldsymbol{\Sigma}' = \mathbf{J}\mathbf{V}\boldsymbol{\Sigma}\mathbf{V}^\top \mathbf{J}^\top$, where $\mathbf{J}$ is the Jacobian matrix of the affine approximation of the projective transformation, and $\mathbf{V}$ denotes the camera view matrix respectively.

## 3.2 GAUSSIAN INITIALIZATION

The vanilla 3DGS (Kerbl et al., 2023) method initializes 3D Gaussians using point clouds generated by Structure-from-Motion (SfM) (Schonberger & Frahm, 2016). However, in the context of endoscopic surgical videos, it is challenging to obtain accurate SfM point clouds due to limited viewpoints, sparse soft tissue textures, and dynamically varying lighting conditions, which hinder precise initialization. To enhance reconstruction quality and stabilize the training process, we perform Gaussian initialization based on prior visual information, including the rgb image $\mathbf{I}$, depth map $\mathbf{D}$, and binary mask $\mathbf{M}$. By incorporating the camera model with known intrinsic and extrinsic parameters, the point cloud can be computed as:

$$\hat{\mathbf{M}} = \bigcap_{i=0}^{T} \mathbf{M}_i, \hat{\mathbf{P}} = \{\hat{\mathbf{D}}\,\mathbf{K}^{-1}\mathbf{T}^{-1}(\hat{\mathbf{I}} \odot (\mathbf{1} - \hat{\mathbf{M}}))\}, \tag{3}$$

where the collected pixels from other frames are added to frame 0 to construct the refined image $\hat{\mathbf{I}}$, depth map $\hat{\mathbf{D}}$, and mask $\hat{\mathbf{M}}$. $\mathbf{M}_i$ denotes the binary mask of frame $i$, with a value of 1 indicating occlusion (*i.e.*, surgical instrument) and 0 indicating visible tissue. The intersection $\hat{\mathbf{M}}$ identifies pixels that are occluded in all frames. Subtracting this result from 1 yields a visibility mask highlighting pixels that are visible in at least one frame. These pixels are retained via element-wise multiplication with the refined image $\hat{\mathbf{I}}$ before being projected into 3D space using the depth map $\hat{\mathbf{D}}$ and the inverse of the intrinsic and extrinsic matrices, $\mathbf{K}^{-1}$ and $\mathbf{T}^{-1}$. The resulting point cloud $\hat{\mathbf{P}}$ is then used to initialize the position $\mu$ and color of the 3D Gaussians.

## 3.3 GAUSSIAN DEFORMATION TRACKING

To achieve high-fidelity reconstruction of dynamic surgical scenes, we propose a deformation modeling strategy called GDT. This strategy decouples deformation tracking into two components: learning the deformation from canonical Gaussian to deformed Gaussian based on 3DGS (Kerbl et al., 2023), and enforcing local geometric constraints to regularize deformation trends. The local geometric constraint specifically regulates the modeling of changes in Gaussian position and shape, enabling high flexibility in capturing complex, high-order variations within the scene.

**Basic Deformation Modeling.** The deformation from the canonical Gaussian to the deformed Gaussian is modeled using a set of MLPs, each dedicated to learning a specific component of the transformation. Specifically, $\mathcal{F}_\mu$, $\mathcal{F}_s$, and $\mathcal{F}_q$ are responsible for predicting the offsets of the Gaussian

center position $\delta\boldsymbol{\mu}$, scale $\delta\mathbf{s}$, and rotation $\delta\mathbf{q}$, respectively. Each network takes as input the center position $\boldsymbol{\mu}$ and the timestamp $t$ of the current frame, which are both processed through a positional encoding function $\gamma(\cdot)$ to capture spatial and temporal variations. Notably, the network does not predict the opacity $\alpha$ or the spherical harmonic (SH) coefficients, as these are considered static attributes of each Gaussian and remain unchanged across time. The final deformed Gaussian at time $t$ is then composed as:

$$\mathcal{G}_d = \{\boldsymbol{\mu} + \delta\boldsymbol{\mu}, \ \mathbf{R} + \delta\mathbf{R}, \ \mathbf{S} + \delta\mathbf{S}, \ \alpha, \ \text{SH}\}. \tag{4}$$

**Local Geometric Constraints.** To further regularize the deformation behavior, we introduce local geometric constraints. Specifically, we first apply Farthest Point Sampling (FPS) (Zhang et al., 2023; Eldar et al., 1997) on the canonical point cloud to select representative Gaussian anchor points. Given the set of all canonical Gaussians $\mathcal{G}_c = \{\boldsymbol{\mu}_i\}_{i=1}^N$, FPS selects a subset $\mathcal{A} = \{\boldsymbol{\mu}_{i_j}\}_{j=1}^M$ such that the minimum pairwise distance between anchor points is maximized:

$$\mathcal{A} = \text{FPS}(\mathcal{G}_c), \quad \text{s.t.} \min_{i \neq j} \|\boldsymbol{\mu}_i - \boldsymbol{\mu}_j\|_2 \text{ is maximized.} \tag{5}$$

Then, for each anchor point $\boldsymbol{\mu}_i \in \mathcal{A}$, we construct a local neighborhood by retrieving its $K$-Nearest Neighbors (KNN) (Zhang et al., 2017; 2023) from $\mathcal{G}_c$ based on Euclidean distance:

$$\mathcal{N}(\boldsymbol{\mu}_i) = \text{KNN}(\boldsymbol{\mu}_i, \mathcal{G}_c, K). \tag{6}$$

For each anchor and its neighbors, we extract the center positions $(\boldsymbol{\mu}_c, \boldsymbol{\mu}_d)$, covariance matrices $(\boldsymbol{\Sigma}_c, \boldsymbol{\Sigma}_d)$, and feature embeddings $(\boldsymbol{f}_c, \boldsymbol{f}_d)$ in both the canonical and deformed spaces. Based on this, we impose three deformation consistency losses: position loss, covariance loss, and feature consistency loss, to ensure locally coherent deformation and to preserve both structural and semantic consistency. The detailed design and computation of these losses are provided in Sec. 3.4.

## 3.4 RENDER & OPTIMIZATION

Reconstruction losses and regularization terms jointly guide our method to optimize the parameters of the canonical Gaussian deformation representation, local geometric consistency constraints, and the low-quality region enhancement module. Since surgical instruments in the video are to be removed, we invert the original masks to focus on soft tissue regions, where reconstruction supervision is exclusively applied. Specifically, unlike previous methods that directly compute the loss between Gaussian-rendered images and the reference images, we account for common degradations in real endoscopic videos, such as motion blur and camera shake. Therefore, we apply a lightweight low-quality enhancement (LQE) method, detailed in the Appendix A.5, to the rendered images before computing the reconstruction loss, enabling more robust alignment with the reference images.

**Color Loss and Depth Loss.** For the $i$-th frame, $\hat{\mathbf{C}}_i$ and $\hat{\mathbf{D}}_i$ denote the RGB image and the corresponding depth map obtained from the initial rendering followed by LQE, respectively.

$$\mathcal{L}_{color} = \frac{1}{HW} \sum_{\mathbf{p}} (1 - \mathbf{M}_i(\mathbf{p}))|\mathbf{I}_i(\mathbf{p}) - \hat{\mathbf{C}}_i(\mathbf{p})|, \ \mathcal{L}_{depth} = \frac{1}{HW} \sum_{\mathbf{p}} (1 - \mathbf{M}_i(\mathbf{p}))|\mathbf{D}_i(\mathbf{p}) - \hat{\mathbf{D}}_i(\mathbf{p})|, \tag{7}$$

where $\mathbf{M}_i$ is the binary tool mask that filters tool pixels, and $H$, $W$ are the image height and width.

**Local Geometric Loss.** Due to the limited viewpoint variations in endoscopic surgical videos, local geometric deformation fields often suffer from severe under-constrained problems, leading to distortion in the transformation from canonical Gaussians to deformed Gaussians. To tackle this challenge, we introduce a local geometric constraint to ensure that neighboring Gaussian primitives exhibit similar deformation behavior. Specifically, we impose local geometric losses on Gaussian position $\mu$, covariance matric $\boldsymbol{\Sigma}$, and point features to enforce consistency of deformation within local regions. As described in Sec. 3.3, each sampled Gaussian $\mathcal{G}_i$ is paired with its $K$ nearest neighbors in the canonical space. We then compute local losses on position, covariance, and feature consistency across both canonical and deformed representations to achieve fine-grained deformation optimization. To enforce structural consistency between domains, we define a set of alignment losses: $\mathcal{L}_{pos}$, $\mathcal{L}_{cov}$, and $\mathcal{L}_{feat}$, which measure domain-wise discrepancy in Gaussian position, covariance, and feature spaces, respectively. Each loss takes the form:

$$\mathcal{L}_x = \sum_{i=1}^N \sum_{k=1}^K \left\| \text{dist}(x_c^{(i)}, x_c^{(k)}) - \text{dist}(x_d^{(i)}, x_d^{(k)}) \right\|_1, \quad x \in \{\boldsymbol{\mu}, \boldsymbol{\Sigma}, f\}, \tag{8}$$

where $\mathcal{L}_{pos} = \mathcal{L}_{\boldsymbol{\mu}}$, $\mathcal{L}_{cov} = \mathcal{L}_{\boldsymbol{\Sigma}}$, and $\mathcal{L}_{feat} = \mathcal{L}_f$. Specifically, $\boldsymbol{\mu}^{(i)}$, $\boldsymbol{\Sigma}^{(i)}$, and $f^{(i)}$ represent the center coordinate, covariance matrix, and encoded feature of the $i$-th Gaussian, respectively. The

subscripts $c$ and $d$ indicate quantities in the canonical and deformed spaces. $\text{dist}(\cdot, \cdot)$ denotes the Euclidean distance. These losses encourage the relative pairwise distances in the source and target domains to remain consistent across multiple feature levels.

**Total Loss.** We combine reconstruction loss $\mathcal{L}_{rec}$ and local geometric loss $\mathcal{L}_{geo}$ to optimize the dynamic 3D Gaussian representation. Additionally, to ensure completeness in the reconstruction of global structures, we incorporate SSIM loss (Li et al., 2022) to enforce structural similarity between the rendered image and the ground-truth image. The relative importance of each loss term is balanced using a set of hyperparameters, and the final optimization objective can be represented as follows:

$$\mathcal{L}_{total} = \underbrace{(\mathcal{L}_{color} + \lambda_1 \mathcal{L}_{ssim} + \lambda_2 \mathcal{L}_{depth})}_{\mathcal{L}_{rec}} + \underbrace{(\lambda_3 \mathcal{L}_{pos} + \lambda_4 \mathcal{L}_{cov} + \lambda_5 \mathcal{L}_{feat})}_{\mathcal{L}_{geo}}. \tag{9}$$

The pseudo-code for the overall method can be found in Appendix A.1.

## 4 EXPERIMENTS

### 4.1 EXPERIMENT SETTINGS

**Datasets and Evaluation.** We conduct experiments on two publicly available surgical video datasets: EndoNeRF (Wang et al., 2022) and StereoMIS (Hayoz et al., 2023). The EndoNeRF dataset is collected from DaVinci robotic (Bodner et al., 2005) surgery video clips, with each clip having a resolution of 512×640 and a frame rate of 15fps. The EndoNeRF dataset includes complex scenes with dynamic scene deformations and tool occlusions, accurately reflecting the various visual and geometric challenges encountered during surgery. Following previous studies, we select two of the most challenging surgical scenarios: *pulling* and *cutting* for evaluation. On the other hand, the StereoMIS dataset is a stereo endoscopic video dataset captured from in-vivo porcine subjects, showcasing diverse anatomical structures and significant tissue deformations, thus presenting more complex scenarios. We select two clips from StereoMIS dataset, which feature a greater diversity of anatomical structures compared to the EndoNeRF dataset. To comprehensively evaluate reconstruction performance, we employ several commonly used quantitative metrics, including Peak Signal-to-Noise Ratio (PSNR) (Sara et al., 2019), Structural Similarity Index Measure (SSIM) (Sara et al., 2019) and Learned Perceptual Image Patch Similarity (LPIPS) (Zhang et al., 2018). We also report rendering speed in frames per second (FPS) and GPU memory usage. For more details, please refer to Appendix A.3.

**Implementation details.** In our implementation, we empirically set the number of initialized points to 30,000. Following prior works, we split the surgical video frames of each scene into a 7:1 training/testing ratio ratio. During training, one frame is randomly selected in each iteration, and all scenes are trained for 40,000 iterations. The Adam optimizer is used with an initial learning rate of $1.6 \times 10^{-3}$, and the remaining training parameters follow the original 3DGS (Kerbl et al., 2023) settings. All experiments are conducted on a single RTX 4090 GPU. For more details, please refer to Appendix A.2.

### 4.2 COMPARISON WITH PRIOR WORKS

We compare LGR with several state-of-the-art SSR methods, including EndoNeRF (Wang et al., 2022), EndoSurf (Zha et al., 2023), LerPlane (Yang et al., 2023), EndoGaussian (Liu et al., 2024b), and SurgicalGaussian (Xie et al., 2024). The quantitative results are summarized in Table 1. Apparently, LGR outperforms all other methods across all reconstruction evaluation metrics and demonstrates superior reconstruction performance while maintaining lower computational overhead. Specifically, LGR achieves the best PSNR values of 39.201 and 38.401 on the EndoNeRF-Pulling and Cutting video clips (Wang et al., 2022), respectively, improving by 0.418 and 0.114 compared to the second-best methods. In terms of perceptual quality measured by LPIPS, which better reflects human visual perception, LGR achieves an average improvement of over 50% on both the EndoNeRF-Pulling and EndoNeRF-Cutting video clips. On the StereoMIS (Hayoz et al., 2023) dataset, LGR also demonstrates consistent superiority, achieving the best LPIPS scores of 0.065 and 0.047 on the S1 and S2 clips, respectively. Additionally, LGR shows improvements in SSIM and PSNR metrics.

We further provide qualitative visual comparisons, as shown in Figure 3. By examining the enlarged regions of interest, LGR achieves the most accurate reconstruction results, which are consistent with

Table 1: Quantitative evaluation on the EndoNeRF (Wang et al., 2022) and StereoMIS (Hayoz et al., 2023) dataset. We report the PSNR↑, SSIM↑, and LPIPS↓ scores. The Avg.FPS↑ and Avg.GPU memory↓ are also provided. The optimal and suboptimal results are shown in **bolded** and underlined respectively.

| Methods | EndoNeRF-Pulling | | | EndoNeRF-Cutting | | | Avg. FPS↑ | Avg. GPU↓ |
|---|---|---|---|---|---|---|---|---|
| | PSNR↑ | SSIM↑ | LPIPS↓ | PSNR↑ | SSIM↑ | LPIPS↓ | | |
| EndoNeRF (Wang et al., 2022) | 34.217 | 0.938 | 0.160 | 34.186 | 0.932 | 0.151 | 0.04 | 15GB |
| EndoSurf (Zha et al., 2023) | 35.004 | 0.956 | 0.120 | 34.981 | 0.953 | 0.106 | 0.05 | 17GB |
| LerPlane (Yang et al., 2023) | 36.241 | 0.950 | 0.102 | 35.580 | 0.955 | 0.101 | 1.02 | 20GB |
| EndoGaussian (Liu et al., 2024b) | 37.308 | 0.958 | 0.070 | 38.287 | 0.962 | 0.058 | **160** | **2GB** |
| SurgicalGaussian (Xie et al., 2024) | 38.783 | 0.970 | 0.049 | 37.505 | 0.961 | 0.062 | 80 | 4GB |
| **LGR (Ours)** | **39.201** | **0.972** | **0.025** | **38.401** | **0.969** | **0.022** | 150 | 4GB |

| Methods | StereoMIS-S1 | | | StereoMIS-S2 | | | Avg. FPS↑ | Avg. GPU↓ |
|---|---|---|---|---|---|---|---|---|
| | PSNR↑ | SSIM↑ | LPIPS↓ | PSNR↑ | SSIM↑ | LPIPS↓ | | |
| EndoNeRF(Wang et al., 2022) | 28.694 | 0.783 | 0.279 | 27.738 | 0.712 | 0.345 | 0.04 | 15GB |
| EndoSurf (Zha et al., 2023) | 29.660 | 0.853 | 0.204 | 28.941 | 0.820 | 0.248 | 0.05 | 17GB |
| LerPlane (Yang et al., 2023) | 29.441 | 0.822 | 0.206 | 28.852 | 0.793 | 0.254 | 1.02 | 20GB |
| EndoGaussian (Liu et al., 2024b) | 29.024 | 0.805 | 0.213 | 26.174 | 0.728 | 0.295 | **160** | **2GB** |
| SurgicalGaussian (Xie et al., 2024) | 31.496 | 0.890 | 0.145 | 31.668 | 0.893 | 0.135 | 80 | 4GB |
| **LGR (Ours)** | **32.444** | **0.919** | **0.065** | **32.273** | **0.924** | **0.047** | 150 | 4GB |

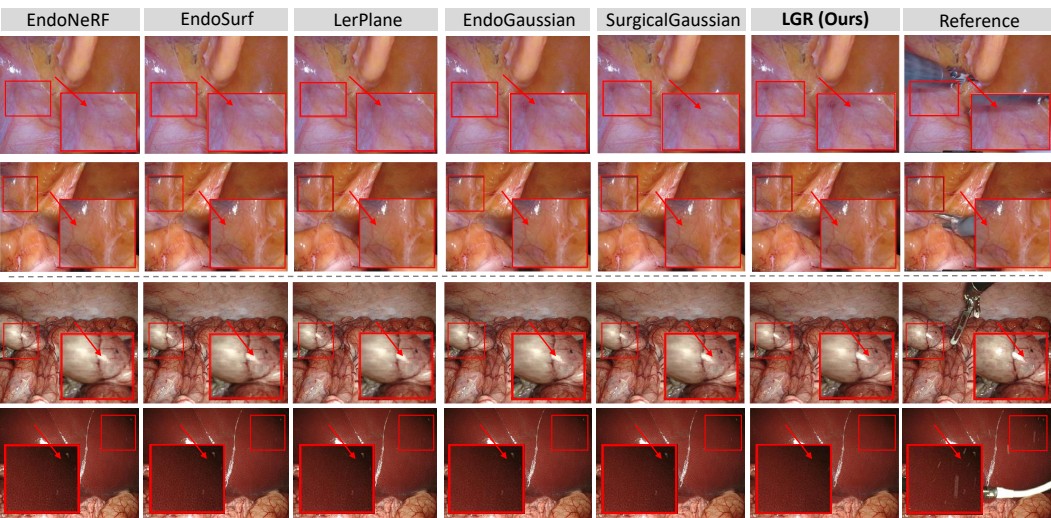

Figure 3: Qualitative evaluation on the EndoNeRF (Wang et al., 2022) and StereoMIS (Hayoz et al., 2023) dataset.

the quantitative evaluations. In contrast, other methods often suffer from texture blurring or loss in dynamic regions of the scene. Specifically, EndoNeRF (Wang et al., 2022) encodes both geometric and appearance changes within a single MLP, limiting its ability to model complex non-rigid deformations. LerPlane (Yang et al., 2023) fixes sampling points on regular grid nodes, which reduces adaptability to spatially localized, nonlinear motion. EndoSurf (Zha et al., 2023) introduces smoothness constraints to improve temporal consistency, but it lacks independent modeling of appearance and geometry, often leading to over-smoothed structures and reduced spatial fidelity. In comparison, 3DGS-based methods such as EndoGaussian (Liu et al., 2024b) and SurgicalGaussian (Xie et al., 2024) have demonstrated improvements in rendering efficiency. However, EndoGaussian uses low-rank tensor feature planes to encode Gaussian deformation fields, which limits its ability to capture complex scene dynamics. SurgicalGaussian (Xie et al., 2024) incorporates local regularization to enhance deformation modeling, but the constraints are applied globally across the entire Gaussian point cloud, resulting in overly rigid structures, increased computational overhead, and reduced flexibility in modeling. In contrast, LGR enforces geometric deformation constraints within the sampled Gaussian points and their local neighborhoods, and integrates a low-quality enhancement module after rendering, leading to more accurate reconstruction in complex surgical scenes. More results and videos are available in Appendix A.4 and the supplementary materials.

Table 2: Ablation study on LGC.

| Model | EndoNeRF-Pulling | | | EndoNeRF-Cutting | | |
|---|---|---|---|---|---|---|
| | PSNR↑ | SSIM↑ | LPIPS↓ | PSNR↑ | SSIM↑ | LPIPS↓ |
| W/O LGC | 38.911 | 0.967 | 0.034 | 38.072 | 0.960 | 0.027 |
| W/ LGC (ours) | **39.201** | **0.972** | **0.025** | **38.401** | **0.969** | **0.022** |

Table 3: Ablation study on LQE.

| Model | StereoMIS-S1 | | | StereoMIS-S2 | | |
|---|---|---|---|---|---|---|
| | PSNR↑ | SSIM↑ | LPIPS↓ | PSNR↑ | SSIM↑ | LPIPS↓ |
| W/O LQE | 32.199 | 0.908 | 0.085 | 32.134 | 0.918 | 0.058 |
| W/ LQE (ours) | **32.444** | **0.919** | **0.065** | **32.273** | **0.924** | **0.047** |

Table 4: Ablation study on FPS Numbers.

| Number | EndoNeRF-Pulling | | | EndoNeRF-Cutting | | |
|---|---|---|---|---|---|---|
| | PSNR↑ | SSIM↑ | LPIPS↓ | PSNR↑ | SSIM↑ | LPIPS↓ |
| 512 | 38.991 | 0.968 | 0.027 | 38.200 | 0.965 | 0.026 |
| 2048 | 39.201 | 0.972 | 0.025 | 38.401 | 0.969 | 0.022 |
| 4096 | 39.279 | 0.974 | 0.025 | 38.414 | 0.971 | 0.021 |

Table 5: Ablation study on KNN Numbers.

| Number | EndoNeRF-Pulling | | | EndoNeRF-Cutting | | |
|---|---|---|---|---|---|---|
| | PSNR↑ | SSIM↑ | LPIPS↓ | PSNR↑ | SSIM↑ | LPIPS↓ |
| 50 | 39.075 | 0.970 | 0.027 | 38.259 | 0.966 | 0.024 |
| 90 | 39.201 | 0.972 | 0.025 | 38.401 | 0.969 | 0.022 |
| 150 | 39.136 | 0.970 | 0.026 | 38.302 | 0.968 | 0.022 |

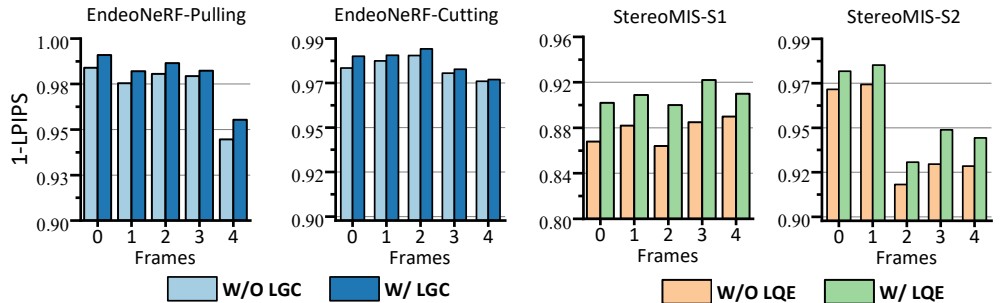

Figure 4: Quantitative evaluation on the EndoNeRF (Wang et al., 2022) and StereoMIS (Hayoz et al., 2023) datasets for analyzing the effect of the LGC and LQE. We randomly select 5 testing images from each scenario and compare the LPIPS scores, all of which show improvements. The overall average results across all test images are reported in Table 2 and Table 3.

## 4.3 ABLATION STUDIES

**Effect of LGC and LQE.** We evaluate the effectiveness of the Local Geometric Constraints (LGC) and Low Quality Enhancement (LQE) modules through comprehensive ablation studies. As shown in Table 2, the removal of the LGC module results in a noticeable decline in PSNR and SSIM, along with a significant increase in LPIPS, indicating degraded reconstruction quality and perceptual consistency. This highlights the crucial role of LGC in promoting structurally coherent motion and improving geometric representation. Similarly, Table 3 demonstrates that excluding the LQE module significantly compromises performance, especially on datasets with a high proportion of low-quality frames such as StereoMIS-S1 and StereoMIS-S2. To further illustrate this, Figure 4 compares LPIPS scores across five randomly selected test images, consistently showing lower values when both modules are present. Moreover, visual comparisons in Figure 5 reveal enhanced edge and texture details in the enlarged regions after integrating LGC and LQE. These observations align with the quantitative improvements, confirming the effectiveness of LGC and LQE in enhancing dynamic scene reconstruction and preserving fine-grained visual details.

**Effect of FPS and KNN Numbers.** We further analyze the impact of two hyperparameters in the LGC module: the number of FPS anchor points and the number of K-Nearest Neighbors (KNN) per anchor. As shown in Table 4, using too few FPS points (*e.g.*, 512) leads to under-constrained deformation and reduced reconstruction quality, while using too many (*e.g.*, 4096) brings only minor performance gains at the cost of increased computational overhead. Similarly, Table 5 shows that a small $K$ limits the expressiveness of local neighborhoods and weakens the modeling of local deformation consistency. Conversely, a large $K$ imposes overly rigid local constraints, which hinders the flexibility of Gaussian deformation, especially in modeling non-rigid and non-uniform tissue motion. In our implementation, setting the number of FPS anchors to 2048 and $K = 90$ effectively balances modeling accuracy and computational efficiency, enhancing scene fidelity while maintaining perceptual consistency.

**Effect of Loss Components.** To evaluate the contribution of each loss component in our framework, we perform a series of ablation experiments on both the EndoNeRF-Pulling and EndoNeRF-Cutting

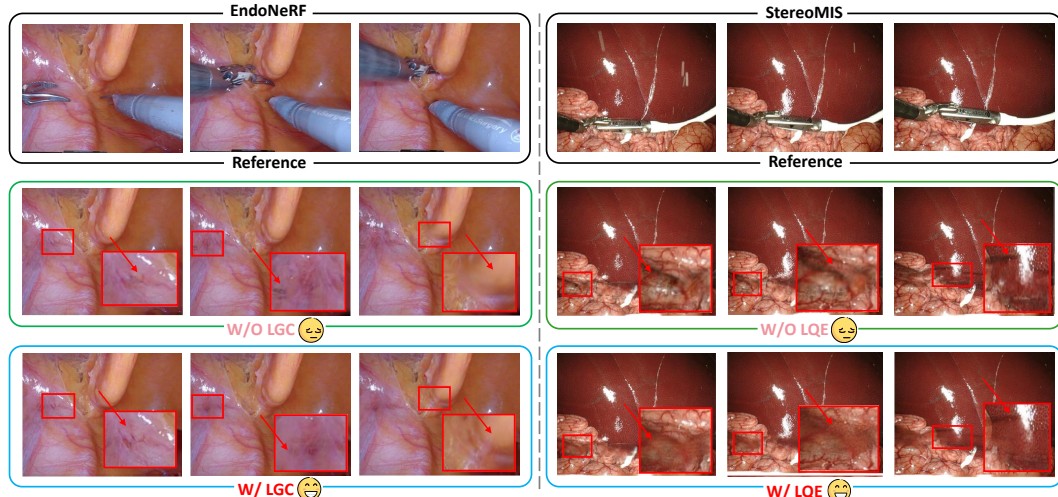

Figure 5: Qualitative evaluation on the EndoNeRF (Wang et al., 2022) and StereoMIS (Hayoz et al., 2023) datasets for analyzing the effect of the LGC and LQE. We select and zoom in on specific detailed regions for comparison.

Table 6: Ablation Study on Loss Components.

| $\mathcal{L}_{color}$ | $\mathcal{L}_{depth}$ | $\mathcal{L}_{ssim}$ | $\mathcal{L}_{pos}$ | $\mathcal{L}_{cov}$ | $\mathcal{L}_{feat}$ | EndoNeRF-Pulling | | | EndoNeRF-Cutting | | |
|---|---|---|---|---|---|---|---|---|---|---|---|
| | | | | | | PSNR ↑ | SSIM ↑ | LPIPS ↓ | PSNR ↑ | SSIM ↑ | LPIPS ↓ |
| ✓ | ✓ | ✗ | ✗ | ✗ | ✗ | 38.452 | 0.960 | 0.051 | 37.201 | 0.956 | 0.042 |
| ✓ | ✓ | ✓ | ✗ | ✗ | ✗ | 38.911 | 0.967 | 0.034 | 38.072 | 0.960 | 0.027 |
| ✓ | ✓ | ✓ | ✓ | ✗ | ✗ | 39.112 | 0.967 | 0.031 | 38.140 | 0.966 | 0.026 |
| ✓ | ✓ | ✓ | ✗ | ✓ | ✗ | 39.118 | 0.970 | 0.026 | 38.327 | 0.968 | 0.024 |
| ✓ | ✓ | ✓ | ✓ | ✓ | ✗ | 39.125 | 0.970 | 0.027 | 38.383 | 0.969 | 0.022 |
| ✓ | ✓ | ✓ | ✓ | ✓ | ✓ | 39.201 | 0.972 | 0.025 | 38.401 | 0.969 | 0.022 |

subsets, as shown in Table 6. Starting from a baseline with only color and depth supervision, we observe significant improvements in all metrics as additional loss terms are incorporated. Specifically, introducing $\mathcal{L}_{ssim}$ and $\mathcal{L}_{pos}$ enhances structural and perceptual consistency, reflected by increased SSIM and decreased LPIPS. Further addition of $\mathcal{L}_{cov}$ and $\mathcal{L}_{feat}$ contributes to finer geometric coherence and feature-level alignment, leading to consistent gains across PSNR, SSIM, and LPIPS. Notably, the full loss configuration achieves the best performance, confirming the complementary roles of these loss components in improving both reconstruction fidelity and perceptual quality. These findings demonstrate the importance of joint supervision from pixel, geometric, and feature spaces for robust modeling of complex surgical scenes.

## 5 CONCLUSION AND LIMITATIONS

**Conclusion.** In this study, we propose a Local Geometric Refinement (LGR) framework for dynamic 3D reconstruction of deformable surgical scenes. LGR integrates Gaussian initialization guided by visual priors, Gaussian deformation tracking under local geometric constraints, and low-quality enhancement. Extensive comparative experiments on public datasets show that LGR improves reconstruction quality in complex surgical environments while maintaining favorable computational efficiency, outperforming existing state-of-the-art methods. Our method has the potential to extend 3D reconstruction technology to practical clinical applications.

**Limitations.** Although LGR has shown promising results in surgical scene reconstruction, its deployment in real medical scenes still faces limitations: i) the implementation of inference in clinical scenes demands high requirements, and the speed of inference needs further improvement; ii) a lack of high-quality video data, constrained by privacy and security concerns, which limits the ability to consider diverse scenarios. Our future work will focus on improving training and inference efficiency, developing lightweight alternatives, and constructing a more comprehensive surgical scene dataset, with the goal of leveraging artificial intelligence more effectively to advance medical research.

## ETHICS STATEMENT

This work does not involve human subjects, sensitive personal data, or applications with direct societal risks. All datasets used are publicly available and have been widely adopted in prior research We therefore believe our study poses no ethical concerns beyond standard practices.

## REPRODUCIBILITY STATEMENT

To facilitate reproducibility, we provide detailed descriptions of dataset usage and model hyperparameters in Sec. 4 and Appendix A. All datasets are publicly available.

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

# A APPENDIX

**Overview.** We thank the reviewers for viewing the appendix. The supplementary includes the following sections:

- Pseudo-code fo Algorithm. (Section A.1).

- Implementation Details. (Section A.2).

- Datastes and Metrics. (Section A.3).

- More Our Results. (Section A.4).

- Details of the Low-Quality Enhancement (LQE) Module. (Section A.5).

- Potential Broader Impacts. (Section A.6).

- Use of Large Language Models (LLMs).(Section A.7).

**The visualization videos will be provided in the supplementary materials package.**

## A.1 PSEUDO-CODE FOR ALGORITHM

---

**Algorithm 1** LGR Framework for Dynamic Surgical Scene Reconstruction

---

1: **Input:** RGB frames $\{I_t\}$, depth maps $\{D_t\}$, masks $\{M_t\}$, camera intrinsics $\mathbf{K}$, extrinsics $\mathbf{T}$
2: **Output:** Reconstructed dynamic 3D Gaussian representation
3: **Gaussian Initialization:**
4:     Aggregate multi-frame data by projecting pixels from all frames onto frame 0
5:     Construct refined image $\hat{\mathbf{I}}$, depth $\hat{\mathbf{D}}$, and mask $\hat{\mathbf{M}}$
6:     Compute visibility mask:$\mathbf{V} = 1 - \hat{\mathbf{M}}$, $\hat{\mathbf{M}} = \bigcap_{i=1}^{T} M_i$
7:     Project visible pixels into 3D: $\hat{\mathbf{P}} = \hat{\mathbf{D}} \cdot \mathbf{K}^{-1}\mathbf{T}^{-1}(\hat{\mathbf{I}} \odot \mathbf{V})$
8:     Initialize Gaussians $\mathcal{G}_c$ from $\hat{\mathbf{P}}$
9: **for** each training iteration **do**
10:     **for** each frame $t$ **do**
11:         **Gaussian Deformation Tracking:**
12:         *// Basic Deformation Modeling*
13:         **for** each Gaussian $\mathcal{G}_i \in \mathcal{G}_c$ **do**
14:             Encode input: $x = \gamma(\boldsymbol{\mu}_i, t)$
15:             Predict offsets: $\delta\mu, \delta s, \delta q = \mathcal{F}_\mu(x), \mathcal{F}_s(x), \mathcal{F}_q(x)$
16:             Update deformed Gaussian: $\mathcal{G}_d^i = \mathcal{G}_c^i + \delta$
17:         **end for**
18:         *// Local Geometric Constraints*
19:         Apply FPS to select anchor set $\mathcal{A} \subset \mathcal{G}_c$
20:         **for** each anchor $i \in \mathcal{A}$ **do**
21:             Retrieve KNN neighbors $\mathcal{N}(\mu_i)$
22:             Compute local consistency losses: $\mathcal{L}_{pos}, \mathcal{L}_{cov}, \mathcal{L}_{feat}$
23:         **end for**
24:         **Rendering and Reconstruction Loss:**
25:             Render image and depth: $(\hat{C}_t, \hat{D}_t) = \text{Render}(\mathcal{G}_d)$
26:             Apply LQE to get enhanced output
27:             Compute pixel-wise losses: $\mathcal{L}_{color}, \mathcal{L}_{depth}, \mathcal{L}_{ssim}$
28:         **Total Loss Computation:**
29:             $\mathcal{L}_{rec} = \mathcal{L}_{color} + \lambda_1\mathcal{L}_{ssim} + \lambda_2\mathcal{L}_{depth}$
30:             $\mathcal{L}_{geo} = \lambda_3\mathcal{L}_{pos} + \lambda_4\mathcal{L}_{cov} + \lambda_5\mathcal{L}_{feat}$
31:             $\mathcal{L}_{total} = \mathcal{L}_{rec} + \mathcal{L}_{geo}$
32:             Backpropagate and update network parameters
33:     **end for**
34: **end for**

---

Table 7: Training Hyperparameters for the LGR

| Parameter | Value | Description |
|---|---|---|
| Iterations | 40000 | Total training steps |
| Initial learning rate $lr_{\text{init}}$ | $1.6 \times 10^{-4}$ | Start learning rate |
| Final learning rate $lr_{\text{final}}$ | $1.6 \times 10^{-6}$ | End learning rate |
| Learning rate delay multiplier $m_{\text{delay}}$ | 0.01 | Scale factor before decay |
| Learning rate maximum steps $S$ | 30000 | Steps for LR decay |
| MLP layers | 8 | MLP depth in BDM |
| MLP width | 256 | MLP width in BDM |
| $\lambda_1, \lambda_2, \lambda_3, \lambda_4, \lambda_5$ | [0.2, 0.001, 1, 200, 0.001] | Loss weights |
| FPS Number | 2048 | Points sampled in LGC |
| KNN Number | 90 | Neighbors used in LGC |

## A.2 IMPLEMENTATION DETAILS

Our proposed LGR method is trained and tested on a single 4090 GPU. The training and testing configurations are summarized below to ensure reproducibility and clarity. Table 7 lists the key hyperparameters used in training the proposed LGR framework, including learning rates, network architecture parameters, and settings for local geometric constraints (LGC) and basic deformation modeling (BDM).

In addition, inspired by Fridovich-Keil et al. (2022), Algorithm 2 provides the pseudo-code for an exponential learning rate scheduler with a warm-up delay mechanism, which controls dynamic learning rate adjustment during training. This approach helps stabilize early-stage optimization while ensuring effective convergence over long training schedules.

---

**Algorithm 2** Exponential Learning Rate Function with Delay

1: **Input:** Initial learning rate $lr_{\text{init}}$, final learning rate $lr_{\text{final}}$, delay steps $s_{\text{delay}}$, delay multiplier $m_{\text{delay}}$, maximum steps $S$
2: **Output:** A function $\text{LR}(s)$ returning the learning rate at step $s$
3: **function** EXPONENTIAL LEARNING RATE($lr_{\text{init}}, lr_{\text{final}}, s_{\text{delay}}, m_{\text{delay}}, S$)
4:     **function** $\text{LR}(s)$
5:         **if** $s < 0$ **or** ($lr_{\text{init}} = 0$ **and** $lr_{\text{final}} = 0$) **then**
6:             **return** $0.0$
7:         **end if**
8:         **if** $s_{\text{delay}} > 0$ **then**
9:             $r \leftarrow \text{clip}(s/s_{\text{delay}}, 0, 1)$
10:            $d \leftarrow m_{\text{delay}} + (1 - m_{\text{delay}}) \cdot \sin(0.5\pi r)$
11:         **else**
12:            $d \leftarrow 1.0$
13:         **end if**
14:         $t \leftarrow \text{clip}(s/S, 0, 1)$
15:         $\ell \leftarrow \exp\left((1 - t) \cdot \log(lr_{\text{init}}) + t \cdot \log(lr_{\text{final}})\right)$
16:         **return** $d \cdot \ell$
17:     **end function**
18:     **return** $\text{LR}(s)$
19: **end function**

---

## A.3 DATASTES AND METRICS

### A.3.1 DATASTES

**We conduct experiments on two publicly available surgical video datasets: EndoNeRF and Stereo-oMIS. Both datasets are licensed under the Creative Commons Attribution-NonCommercial-ShareAlike 4.0 International License (CC BY-NC-SA 4.0), do not involve any privacy violations, and have been properly cited in our work.** These datasets provide realistic, diverse surgical

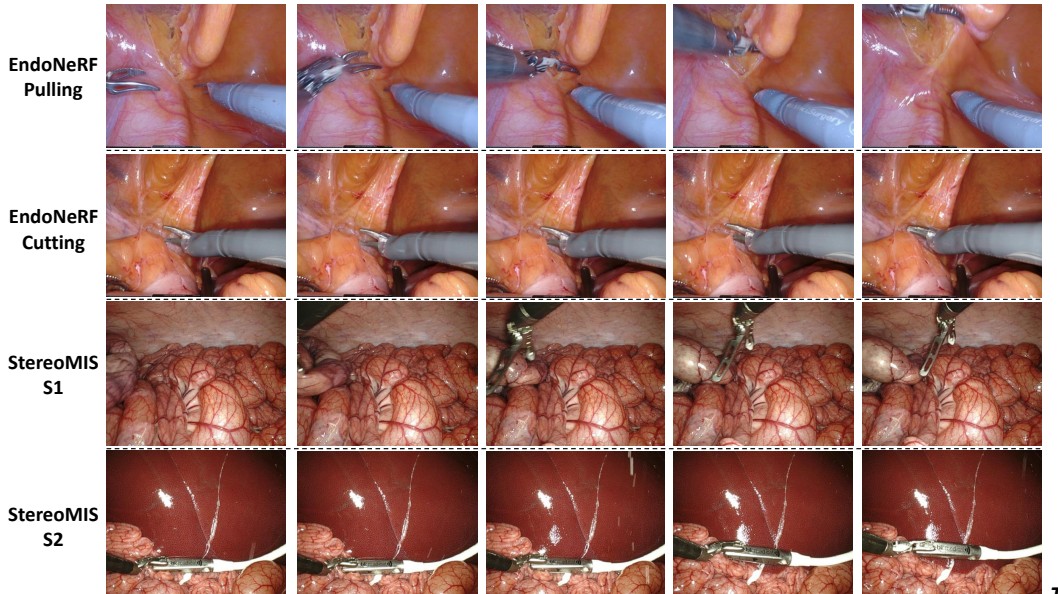

Figure 6: Representative samples from the EndoNeRF and StereoMIS datasets illustrating the diversity of scenes used in our experiments.

Table 8: Comparison of EndoNeRF and StereoMIS Datasets

| Attribute | EndoNeRF | StereoMIS |
|---|---|---|
| Capture Device | Da Vinci Surgical Robot | Da Vinci Xi Surgical Robot |
| Resolution | $640 \times 512$ | $640 \times 512$ (downsampled) |
| Surgical Type | Prostatectomy (Human) | In-vivo Porcine Study |
| Key Challenges | Deformation, Tool Occlusion | Anatomical Diversity, Deformation |
| Additional Data | Depth, Tool Masks | Camera Poses, Kinematics |

scenes and serve as valuable benchmarks for evaluating surgical scene reconstruction methods under real-world challenges.

**EndoNeRF Dataset.** The EndoNeRF dataset is specifically designed for stereo 3D reconstruction tasks. It consists of stereo video clips collected from Da Vinci robotic-assisted prostatectomy surgeries. Each clip has a resolution of $640 \times 512$ and a frame rate of 15 fps. The dataset contains complex surgical scenes with dynamic soft tissue deformation and frequent tool occlusion, accurately reflecting the visual and geometric difficulties encountered in minimally invasive surgery. Following previous studies, we select two of the most challenging scenarios *pulling* and *cutting* for evaluation. In addition to the RGB frames, EndoNeRF also provides estimated depth maps and manually annotated tool masks to support supervised learning and detailed evaluation.

**StereoMIS Dataset.** The StereoMIS dataset is a large-scale stereo endoscopic video dataset acquired from in-vivo porcine experiments using the Da Vinci Xi surgical system. It consists of 11 stereo sequences recorded during real surgical procedures, with each sequence showcasing diverse anatomical structures and substantial tissue deformation. The original resolution is $1280 \times 1024$ at 60 fps, downsampled to $640 \times 512$ for compatibility. Each sequence includes synchronized stereo pairs along with forward kinematics and camera pose data, making it suitable for evaluating both scene reconstruction and camera tracking. We select two representative clips from StereoMIS that exhibit more anatomical diversity compared to EndoNeRF.

These two datasets provide complementary scenarios for comprehensive evaluation. As summarized in Table 8, the EndoNeRF dataset emphasizes high-fidelity stereo reconstruction in human surgical

environments, whereas the StereoMIS dataset captures more anatomically diverse and dynamically deforming tissue structures from preclinical studies. Each dataset introduces distinct challenges in terms of anatomical complexity, tissue dynamics, and camera motion, thereby enabling thorough assessment of non-rigid reconstruction approaches across varied conditions. Representative samples from both datasets are shown in Figure 6, illustrating their differences in structural variability and visual characteristics. Together, these datasets constitute a robust and diverse benchmark for evaluating reconstruction algorithms under both structural and perceptual challenges.

### A.3.2 METRICS

**Peak Signal-to-Noise Ratio (PSNR).** PSNR is a widely used metric for evaluating the pixel-wise fidelity between a reconstructed image and its ground truth counterpart. It is particularly relevant in surgical scene reconstruction tasks, where precise intensity recovery is essential for preserving anatomical details. PSNR is defined based on the Mean Squared Error (MSE) between the predicted and ground truth images. Given an image of resolution $H \times W$, PSNR is computed as:

$$\text{PSNR} = 10 \cdot \log_{10}\left(\frac{MAX^2}{\text{MSE}}\right), \quad \text{where} \quad \text{MSE} = \frac{1}{HW}\sum_{i=1}^{H}\sum_{j=1}^{W}\left(I(i,j) - \hat{I}(i,j)\right)^2, \quad (10)$$

where $I$ and $\hat{I}$ denote the ground truth and reconstructed images, respectively, and $MAX$ is the maximum possible pixel value (typically 1.0 or 255). Higher PSNR values indicate better reconstruction accuracy in terms of low-level pixel similarity, which is especially important for recovering fine textures in surgical scenes.

**Structural Similarity Index Measure (SSIM).** SSIM measures the perceptual similarity between two images by evaluating their luminance ($l$), contrast ($c$), and structural consistency ($s$). It is particularly suitable for surgical scene reconstruction, where preserving spatial structure and tissue morphology is critical. Given image patches $x$ and $y$, SSIM is defined as:

$$\text{SSIM}(x,y) = \frac{(2\mu_x\mu_y + C_1)(2\sigma_{xy} + C_2)}{(\mu_x^2 + \mu_y^2 + C_1)(\sigma_x^2 + \sigma_y^2 + C_2)}, \quad (11)$$

where $\mu_x$, $\mu_y$ are the local means, $\sigma_x^2$, $\sigma_y^2$ are the variances, and $\sigma_{xy}$ is the covariance of patches $x$ and $y$. $C_1$ and $C_2$ are constants to stabilize the division. SSIM ranges from 0 to 1, with higher values indicating better structural consistency—a key property in reconstructing deformable surgical scene.

**Learned Perceptual Image Patch Similarity (LPIPS).** LPIPS evaluates perceptual similarity by comparing deep feature activations extracted from pretrained convolutional neural networks, such as AlexNet or VGG. It provides a measure of high-level perceptual closeness, which aligns well with human visual judgment—an important consideration in assessing visual quality in surgical scene rendering. The LPIPS score between two images $x$ and $y$ is computed as:

$$\text{LPIPS}(x,y) = \sum_l \frac{1}{H_l W_l} \sum_{h,w} \left\| w_l \odot \left(f_l^x(h,w) - f_l^y(h,w)\right)\right\|_2^2, \quad (12)$$

where $f_l^x$ and $f_l^y$ are the feature maps of images $x$ and $y$ at layer $l$, $w_l$ is a learned weight vector, and $\odot$ denotes element-wise multiplication. Lower LPIPS values indicate greater perceptual similarity. In surgical scene reconstruction, LPIPS helps evaluate whether reconstructed images are visually realistic and consistent with human perception, beyond just pixel-level accuracy.

### A.4 MORE OUR RESULTS

To further demonstrate the effectiveness of our proposed LGR framework, we present additional quantitative and qualitative results. Figure 7 compares the reconstruction performance of LGR with existing methods on the EndoNeRF-Pulling and StereoMIS-S1 test datasets, evaluated using PSNR and SSIM metrics. Extended visual comparisons on the EndoNeRF-Pulling, EndoNeRF-Cutting, StereoMIS-S1, and StereoMIS-S2 datasets are provided in Figure 8 and Figure 9. In light of the availability of corresponding open-source code, we have included quantitative comparisons with Endo-4DGS (Huang et al., 2024), Deform3DGS (Yang et al., 2024b), and EH-SurGS (Shan et al.,

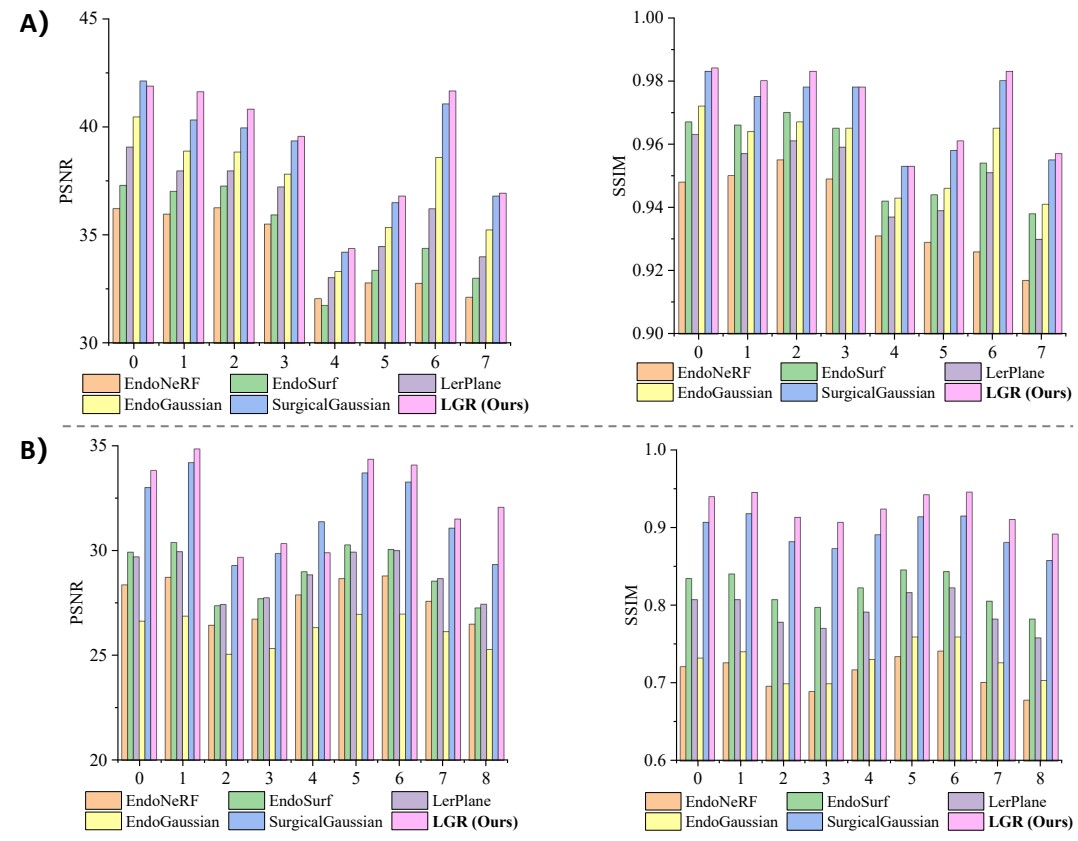

Figure 7: Quantitative comparison between LGR and comparison methods. **A)** PSNR and SSIM comparisons on individual frames from the EndoNeRF-Pulling test dataset. **B)** PSNR and SSIM comparisons on individual frames from the StereoMIS-S1 test dataset.

Table 9: Quantitative evaluation on the EndoNeRF (Wang et al., 2022) and Hamlyn (Mountney et al., 2010) dataset. We report the PSNR↑, SSIM↑, and LPIPS↓ scores.

| Methods | EndoNeRF-Pulling | | | EndoNeRF-Cutting | | |
|---|---|---|---|---|---|---|
| | PSNR↑ | SSIM↑ | LPIPS↓ | PSNR↑ | SSIM↑ | LPIPS↓ |
| Deform3DGS (Yang et al., 2024b) | 37.816 | 0.958 | 0.062 | 36.918 | 0.958 | 0.065 |
| Endo-4DGS (Huang et al., 2024) | 37.183 | 0.955 | 0.072 | 36.132 | 0.951 | 0.054 |
| EH-SurGS (Shan et al., 2025) | 38.205 | 0.960 | 0.061 | 38.057 | 0.963 | 0.057 |
| SurgicalGaussian (Xie et al., 2024) | 38.783 | 0.970 | 0.049 | 37.505 | 0.961 | 0.062 |
| **LGR (Ours)** | **39.201** | **0.972** | **0.025** | **38.401** | **0.969** | **0.022** |

| Methods | Hamlyn-1 | | | Hamlyn-2 | | |
|---|---|---|---|---|---|---|
| | PSNR ↑ | SSIM↑ | LPIPS ↓ | PSNR ↑ | SSIM↑ | LPIPS↓ |
| Deform3DGS (Yang et al., 2024b) | 29.946 | 0.930 | 0.139 | 31.902 | 0.947 | 0.131 |
| Endo-4DGS (Huang et al., 2024) | 27.506 | 0.921 | 0.158 | 32.111 | 0.948 | 0.112 |
| EH-SurGS (Shan et al., 2025) | 33.842 | 0.956 | 0.077 | 35.413 | 0.964 | 0.083 |
| SurgicalGaussian (Xie et al., 2024) | 31.436 | 0.934 | 0.125 | 33.161 | 0.948 | 0.122 |
| **LGR (Ours)** | **34.401** | **0.958** | **0.072** | **36.466** | **0.968** | **0.074** |

2025). Furthermore, we have conducted experiments on the Hamlyn dataset. The experimental results, summarized in Table 9, demonstrate that our method achieves excellent reconstruction performance. Additional video results of reconstructed surgical scenes are available in the supplementary material to better showcase the spatiotemporal fidelity of our method. The visualization videos will be provided in the supplementary materials package.

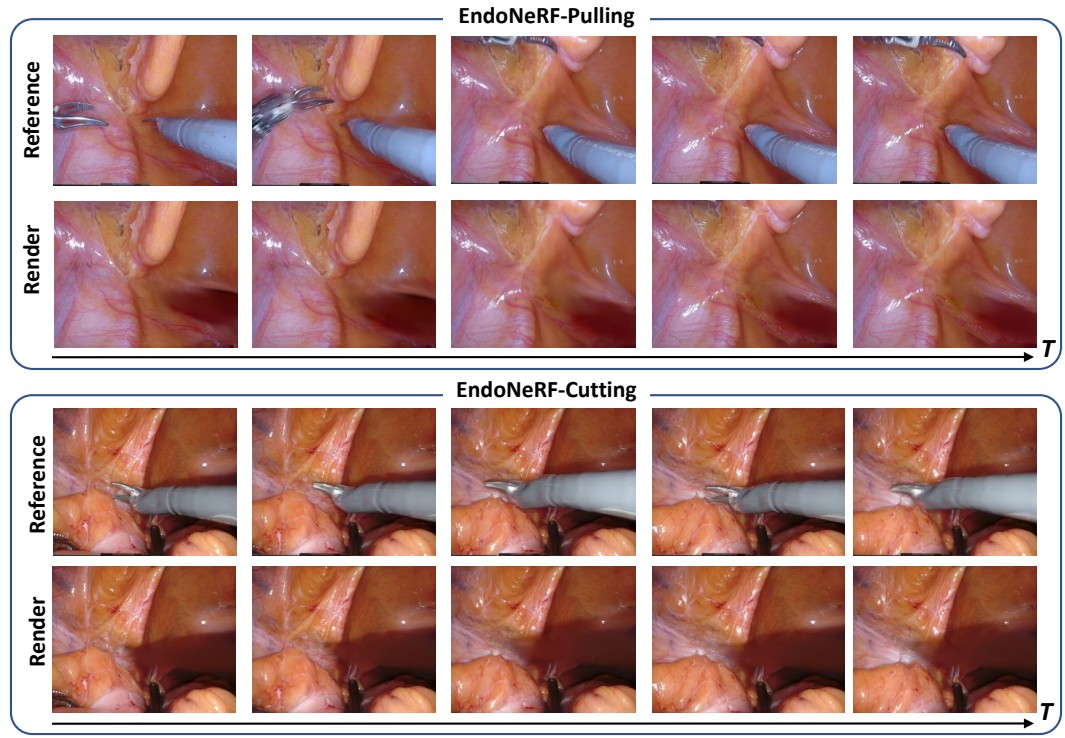

Figure 8: Additional qualitative results of LGR on the EndoNeRF-Pulling and EndoNeRF-Cutting datasets.

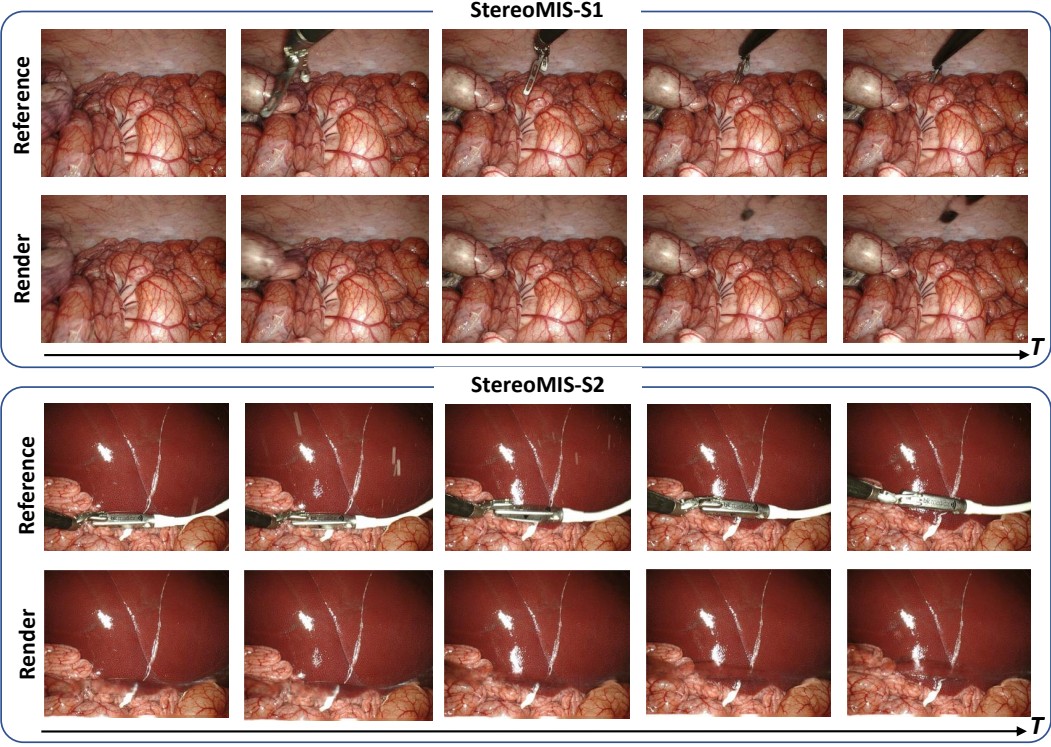

Figure 9: Additional qualitative results of LGR on the StereoMIS-S1 and StereoMIS-S2 datasets.

### A.5 DETAILS OF THE LOW-QUALITY ENHANCEMENT (LQE) MODULE

We provide a detailed description of the structure and workflow of the Low-Quality Enhancement (LQE) module in the form of pseudocode below. Our design is mainly inspired by the contributions of Restormer (Zamir et al., 2022) and RSDFormer (Song et al., 2023) in low-level vision tasks. Due to the limited size of surgical scene datasets in this project, we replace the main network architecture from Transformer to CNN. The pseudocode 3 is as follows:

---

**Algorithm 3** Workflow of Low-Quality Enhancement (LQE)

---

1: **Input:** x (3-channel RGB or 1-channel depth image)
2: **Output:** Enhanced image y
3: **function** LQE(x)
4:     *// Embedding*
5:     **if** channel(x) == 3 **then**
6:         $x_1 \leftarrow$ Conv2d(x, $3 \to 32$, k=3)                    ▷ Embed RGB to feature space
7:     **else**
8:         $x_1 \leftarrow$ Conv2d(x, $1 \to 32$, k=3)                    ▷ Embed Depth to feature space
9:     **end if**

10:     *// Encoder Path*                    ▷ ConvBlock = $2\times$(dsconv + LayerNorm + ReLU)
11:     $enc_1 \leftarrow$ ConvBlock($x_1$, $32 \to 32$)
12:     $down_1 \leftarrow$ Downsample($enc_1$)                    ▷ Conv($32 \to 16$) + PixelUnshuffle($\times 4$)
13:     $enc_2 \leftarrow$ ConvBlock($down_1$, $64 \to 64$)
14:     $down_2 \leftarrow$ Downsample($enc_2$)                    ▷ Conv($64 \to 32$)+PixelUnshuffle($\times 4$)
15:     $enc_3 \leftarrow$ ConvBlock($down_2$, $128 \to 128$)
16:     $down_3 \leftarrow$ Downsample($enc_3$)                    ▷ Conv($128 \to 64$) + PixelUnshuffle($\times 4$)
17:     $bottleneck \leftarrow$ ConvBlock($down_3$, $256 \to 256$)

18:     *// Decoder Path*
19:     $up_3 \leftarrow$ Upsample($bottleneck$)                    ▷ Conv($256 \to 512$) + PixelShuffle($\div 4$)
20:     $cat_3 \leftarrow$ Concat($up_3$, $enc_3$)                    ▷ Concatenate on channel dim
21:     $dec_3 \leftarrow$ ConvBlock(Conv1x1($cat_3$, $256 \to 128$), $128 \to 128$) ▷ Reduce channels & decode

22:     $up_2 \leftarrow$ Upsample($dec_3$)                    ▷ Conv($128 \to 256$) + PixelShuffle($\div 4$)
23:     $cat_2 \leftarrow$ Concat($up_2$, $enc_2$)
24:     $dec_2 \leftarrow$ ConvBlock(Conv1x1($cat_2$, $128 \to 64$), $64 \to 64$)

25:     $up_1 \leftarrow$ Upsample($dec_2$)                    ▷ Conv($64 \to 128$) + PixelShuffle($\div 4$)
26:     $cat_1 \leftarrow$ Concat($up_1$, $enc_1$)
27:     $dec_1 \leftarrow$ ConvBlock(Conv1x1($cat_1$, $64 \to 32$), $32 \to 32$)

28:     *// Refinement & Output*
29:     $refine \leftarrow$ ConvBlock($dec_1$, $32 \to 32$)
30:     **if** channel(x) == 3 **then**
31:         $output \leftarrow$ Conv2d($refine$, $32 \to 3$, $k = 3$)
32:     **else**
33:         $output \leftarrow$ Conv2d($refine$, $32 \to 1$, $k = 3$)
34:     **end if**

35:     *// Residual Connection*
36:     $y \leftarrow output + x$
37:     **return** $y$
38: **end function**

---

### A.6 POTENTIAL BROADER IMPACTS

The proposed LGR method demonstrates significant potential in modeling and tracking non-rigid deformations in surgical scenes. Beyond its technical contributions, LGR may have broader impacts across various domains:

1. **Enhancing Precision and Safety in Image-Guided Surgeries:** By integrating local geometric constraints, the proposed LGR method effectively captures non-rigid deformations, thereby improving the accuracy of endoscopic tracking and surgical scene reconstruction. This advancement assists surgeons in navigating instruments with greater precision, potentially reducing inadvertent damage to healthy tissues and enhancing overall surgical outcomes.

2. **Advancing Medical Education and Virtual Reality Training:** High-fidelity modeling of soft tissue deformations is crucial for developing realistic virtual surgical training systems. LGR's capability to simulate authentic tissue behavior enriches medical education by providing immersive and interactive training platforms, allowing medical professionals to practice complex procedures in a risk-free environment.

3. **Fostering Cross-Disciplinary Technological Innovations:** The proficiency of LGR in handling non-rigid deformations extends its applicability to fields such as robotics, augmented reality (AR), and virtual reality (VR). For instance, in robotic surgery, precise tissue tracking can facilitate higher levels of automation, while in AR/VR applications, realistic deformation modeling enhances user immersion, thereby driving innovation across multiple technological domains.

## A.7 USE OF LARGE LANGUAGE MODELS (LLMS)

We used Large Language Models (LLMs), specifically OpenAI's ChatGPT (GPT-4o/5), as an assistive tool during the preparation of this paper. The LLMs were used for language polishing, grammar refinement, and improving readability of the text. All technical content, data interpretation, and scientific contributions were generated entirely by the authors. The authors take full responsibility for the correctness, originality, and integrity of the content presented in this paper.

