# OpenReview forum: "LGR: Local Geometric Refinement in High-Fidelity Surgical Scene Reconstruction"
_ICLR.cc/2026/Conference — ICLR 2026 Conference Withdrawn Submission_

### Official Review · Reviewer_Vxn7 · 2025-10-26

**Soundness:** 3
**Presentation:** 4
**Contribution:** 3
**Rating:** 6
**Confidence:** 5

**Summary:**

This paper introduces LGR, a 3D Gaussian Splatting framework for semantic and dynamic surgical scene reconstruction. It incorporates two main innovations: (1) LGC that regularize Gaussian neighborhoods for better deformation tracking, and (2) a LQE module that improves perceptual fidelity in degraded surgical frames. The method is evaluated on synthetic and real endoscopic datasets, showing strong LPIPS reductions and improved visual consistency compared to prior NeRF and 3DGS-based methods. The authors emphasize real-time rendering, generalizability to low-quality frames, and perceptual improvements enabled by the proposed modules. I have noticed the open claim, and it is appreciated.

**Strengths:**

1. Addresses a challenging and high-impact domain: dynamic surgical scene reconstruction and thoughtfully designed local geometric constraints: LGC improves non-rigid tracking with minimal overhead.
2. LQE module enhances noisy real surgical data: practical relevance, especially in clinical conditions.
3. I have noticed comprehensive ablations: LGC loss terms and LQE effects are isolated and quantified.
4. Runtime, memory, and perceptual metrics reported: aligns with real-world use cases.

**Weaknesses:**

While the method is well-motivated and empirically strong, several limitations should be noted. First, the potential for metric inflation via LQE remains a concern: it is unclear whether LPIPS, PSNR, and SSIM are computed against raw ground-truth images or LQE-enhanced targets. If both outputs and ground truths are post-processed, reported gains may conflate reconstruction quality with enhancement effects. Second, the supplementary material provides a PowerPoint file rather than a video, why? Moreover, qualitative comparisons across methods show only subtle differences; zoomed-in visualizations or pixel-wise error maps (e.g., LPIPS heatmaps) would better support perceptual claims. Third, although LGR is framed as generalizable, it is tightly coupled to domain-specific priors such as depth maps and surgical tool masks. Its adaptability to non-surgical scenes or monocular input remains unclear, and its robustness to noisy or missing priors is not evaluated. Finally, while the LQE module is shown to improve perceptual fidelity, its architecture and training regime are only briefly described. Further detail is needed to understand whether LQE introduces its own artifacts, how it avoids overfitting, and how it interacts with the rest of the pipeline. Besides, since the LQE is a common method, can the LQE imporve the performance of previous metheds like endonerf / forplane/ surgicalgaussian/endogaussian? Finally, the method relies on pre-calibrated camera poses, yet makes no reference to surgical SLAM systems (e.g., EndoSLAM, EndoGSLAM), which directly address pose estimation in deformable surgical scenes. These are foundational to the assumptions of this paper and must be cited.

**Questions:**

1. Are perceptual metrics (LPIPS, PSNR, SSIM) computed against the raw GT or LQE-enhanced GT? Please clarify.
2. Can the authors provide zoomed-in qualitative comparisons or error heatmaps to support claims of better perceptual quality?
3. Would the method still perform well without explicit depth and mask supervision, such as in monocular or weakly supervised settings?

---

### Official Review · Reviewer_4JnK · 2025-10-28

**Soundness:** 2
**Presentation:** 3
**Contribution:** 2
**Rating:** 2
**Confidence:** 4

**Summary:**

The paper proposed LGR, a framework for adding some deformation constrains to dynamic Gaussian Splatting Pipeline on sugical scene. Specially, an initialization with prior (depth and mask), a Local Geometric Loss and a LQE method for enhancing the rendered image are proposed. Experiments on EndoNeRF and StereoMIS show the effectiveness of the proposed method.

**Strengths:**

- The paper is clear and well-written. The proposed method is easy to follow.
- The paper thoroughly discusses the related work.
- The utilized datasets and experimental settings are introduced in detail.

**Weaknesses:**

1. **Difference on Gaussian Initialization between EndoGaussian?**: The proposed 'Gaussian Initialization' module appears to be similar to the initialization method used in EndoGaussian [1]. It would be beneficial to include a more detailed discussion on the differences between the two approaches.

2. **Clarification for Gaussian Deformation Tracking**:
- The paper's main motivation is to add deformation constraints during reconstruction, which selects a set of anchor Gaussians and tracks their local geometry with their neighbours. The selection process seems less motivated.
- More Visualization results about the local geometry structure of anchor Gaussian and their neighbours during deformation should be add to further prove the effectiveness of the proposed method.
- Existing tracking methods should be discussed. More metrics on "Tracking" to prove the effectiveness.
- Conducting KNN directly via Euclidean distance can select neighboring Gaussians that belong to different semantic objects, which invalidates the local supervision. The performance drop observed in Tab. 5 as the KNN number increases might be because the model is over-regularizing these invalid structures.

3. **Clarification for the proposed LQE method**:
- The proposed method seems an important contribution of the paper, but all details are in the Appendix. The paper should be reorganized for better reading.
- The training details of the LQE module should be clarified. Is it jointly trained with the Gaussians?
- The performance improvement over EndoNeRF is marginal. Is this gain a cumulative effect of all the proposed modules working together? An ablation study featuring the baseline method + the LQE module should be included to further demonstrate its effectiveness.

4. **Experiemtal results**:
- Experiments are only conducted on four scenes. More datasets should be included to prove the effectiveness.
- The improvement of over 50% in LPIPS is not obvious in the current visual results.

[1] EndoGaussian: Real-time Gaussian Splatting for Dynamic Endoscopic Scene Reconstruction.

**Questions:**

Refer to the weakness section.

**Details Of Ethics Concerns:**

No Ethics Concerns.

---

### Official Review · Reviewer_C4RT · 2025-10-30

**Soundness:** 2
**Presentation:** 3
**Contribution:** 1
**Rating:** 2
**Confidence:** 5

**Summary:**

This paper proposes LGR, a framework for high-fidelity dynamic reconstruction of deformable surgical scenes based on 3D Gaussian Splatting (3DGS). While recent 3DGS-based methods have advanced surgical scene reconstruction, they often fail to accurately capture local non-rigid tissue deformations due to the lack of geometric constraints in local Gaussian neighborhoods and are sensitive to low-quality visual artifacts (e.g., blood splashes, motion blur) common in real surgeries. To address these limitations, LGR introduces three key components:1. Gaussian Initialization using multi-frame visual priors (RGB, depth, and tool masks) for robust point cloud seeding. 2. Local Geometric Constraints (LGC) during deformation tracking, which enforce consistency in position, covariance, and feature space among each Gaussian and its K-nearest neighbors, enabling precise modeling of fine-grained local deformations. 3. A Low-Quality Enhancement (LQE) module applied after rendering to refine details degraded by surgical visual artifacts.

**Strengths:**

1. Clear and structured writing: The paper is well-structured, logically coherent, and presents its methodology with sufficient detail.
2. Reasonable motivation: It tries to address two real-world surgical challenges—difficulty in tracking fine local tissue deformations and poor image quality—and proposes targeted solutions.

**Weaknesses:**

1. Limited technical novelty in Gaussian initialization: The proposed Gaussian initialization largely replicates the strategy used in EndoGaussian (Liu et al., 2024b), relying on back-projection of RGB, depth, and mask priors without introducing a fundamentally new approach.
2. Local geometric constraints lack originality: The Local Geometric Constraints (LGC) module employs a standard KNN-based neighborhood consistency formulation—common in many non-rigid deformation frameworks—and does not significantly advance beyond existing practices. Moreover, methods like HexPlane already inherently enforce local smoothness through their structured feature representation, making the added LGC potentially redundant.
3. Low-Quality Enhancement (LQE) appears superficial: The LQE module seems to apply an off-the-shelf CNN-based image enhancement pipeline (inspired by Restormer/RSDFormer) to the rendered output, without clear justification that it is jointly optimized or specifically adapted to the surgical reconstruction task. It may simply be enhancing the ground truth or rendered image as a post-processing step, offering minimal integration with the core 3DGS framework.
4. Incomplete experimental validation: While experiments on EndoNeRF and StereoMIS are provided, the paper omits evaluation on the Hamlyn dataset beyond a brief mention in the appendix. Given that Hamlyn contains more diverse surgical cases (seven cases?) and is widely used in SSR literature, its exclusion weakens the comprehensiveness of the benchmarking and raises questions about generalizability.

**Questions:**

1. To what extent does the Gaussian initialization differ from that of EndoGaussian (Liu et al., 2024b)? Given that both methods rely on back-projecting RGB, depth, and tool mask priors to seed the Gaussian point cloud, what constitutes the technical novelty of this component in LGR?
2. Does the Local Geometric Constraints (LGC) module offer a meaningful advance over existing local consistency strategies? Since KNN-based neighborhood regularization is a well-established technique in non-rigid deformation modeling.
3. How deeply integrated is the Low-Quality Enhancement (LQE) module into the core 3DGS reconstruction pipeline? If LQE is essentially a pre-trained CNN-based post-processing step applied to rendered images (inspired by Restormer/RSDFormer), in what way is it specifically tailored to or jointly optimized with the surgical scene reconstruction task, rather than acting as a generic image enhancer?
4. Why is the Hamlyn dataset not included in the main experimental evaluation? Given that Hamlyn contains multiple diverse surgical cases and is a standard benchmark in surgical scene reconstruction, doesn’t its omission from the primary results limit the assessment of LGR’s generalizability and robustness across varied clinical scenarios?

---

### Official Review · Reviewer_CWyz · 2025-11-01

**Soundness:** 2
**Presentation:** 3
**Contribution:** 2
**Rating:** 2
**Confidence:** 4

**Summary:**

This paper introduces LGR for dynamic reconstruction from the deformable surgical scenes. The key idea is to use a local geometric refinement to accurately track the local non-rigid deformations occurring in the surgical scene. The experiments show that LGR achieves state-of-the-art reconstruction quality, improving over 50 % in LPIPS while maintaining the computational cost.

**Strengths:**

1. The paper presents a good framework to achieve dynamic reconstruction for surgical soft tissues, showing improved performance over previous methods.
2. The proposed local geometric loss is simple yet effective to constrain the neighboring Gaussian primitives with similar deformation, resulting in consistency during deformation optimization.
3. The paper shows comprehensive experiments to include ablation studies for all key designs with careful analysis.

**Weaknesses:**

1. The novelty of the paper is rather limited. The core novelty of the paper to constrain the local consistency is not new and is essentially the same as the as-rigid-as-possible loss/rigidity loss, which has been widely used in the previous 4DGS works [1][2][3]. The experimental comparisons and analysis with these previous methods are also needed to show the improvement of the proposed work.
2. The proposed LQE method also lacks novelty, which seems like a post-processing step to refine/repair the quality of rendered images before computing loss. However, this is not fair to the compared methods without this refinement. Additionally, it remains unclear whether the FPS results reported in Table 1 include the computational cost of this post-processing step.
3. The related work is not sufficient and lacks a lot in 4DGS. As mentioned in 1, the related work needs to include the recent 4DGS works, but not limited to surgical reconstruction with 3DGS or NeRF.
4. Missing comparison with recent 4DGS methods, for example, Surgicalgs[4], MoDGS[5], Instant4D[6], 4DGC[7]
5. The effectiveness of the method is also questionable. As shown in Table 6, it seems that L_ssim accounts for the majority of gains in LPIPS and PSNR, while the proposed LGC-related losses contribute only marginal improvements. Additionally, Table 3 and Table 2 show that the module LQE contributes most to the improvement on LPIPS, which is a post-refinement module and not the core contribution of the paper.


[1] Huang, Yi-Hua, et al. "Sc-gs: Sparse-controlled gaussian splatting for editable dynamic scenes." Proceedings of the IEEE/CVF conference on computer vision and pattern recognition. 2024.

[2] Lei, Jiahui, et al. "Mosca: Dynamic gaussian fusion from casual videos via 4d motion scaffolds." Proceedings of the Computer Vision and Pattern Recognition Conference. 2025.

[3] Wang, Qianqian, et al. "Shape of motion: 4d reconstruction from a single video." Proceedings of the IEEE/CVF International Conference on Computer Vision. 2025.

[4] Chen, Jialei, et al. "Surgicalgs: Dynamic 3d gaussian splatting for accurate robotic-assisted surgical scene reconstruction." International Conference on Medical Image Computing and Computer-Assisted Intervention. Cham: Springer Nature Switzerland, 2025.

[5] Qingming, L. I. U., et al. "MoDGS: Dynamic gaussian splatting from casually-captured monocular videos with depth priors." The Thirteenth International Conference on Learning Representations. 2025.

[6] Luo, Zhanpeng, Haoxi Ran, and Li Lu. "Instant4D: 4D Gaussian Splatting in Minutes." arXiv preprint arXiv:2510.01119 (2025).

[7] Hu, Qiang, et al. "4DGC: Rate-Aware 4D Gaussian Compression for Efficient Streamable Free-Viewpoint Video." Proceedings of the Computer Vision and Pattern Recognition Conference. 2025.

**Questions:**

Please refer to the weaknesses. Overall, I believe the paper is not ready for publication, due to the limited novelty, insufficient experiments, and questionable effectiveness.

---

### Note · Authors · 2025-11-16

I have read and agree with the venue's withdrawal policy on behalf of myself and my co-authors.